# Finding, visualizing, and quantifying latent structure across diverse animal vocal repertoires

Tim Sainburg [1]   Marving Thielk [1]   Timothy Q. Gentner [1]

## Abstract

Animals produce vocalizations that range in complexity from a single repeated call to hundreds of unique vocal elements patterned in sequences unfolding over hours. Characterizing complex vocalizations can require considerable effort and a deep intuition about each species' vocal behavior. Even with a great deal of experience, human characterizations of animal communication can be affected by human perceptual biases. We present a set of computational methods for projecting animal vocalizations into low dimensional latent representational spaces that are directly learned from the spectrograms of vocal signals. We apply these methods to diverse datasets from over 20 species, including humans, bats, songbirds, mice, cetaceans, and nonhuman primates. Latent projections uncover complex features of data in visually intuitive and quantifiable ways, enabling high-powered comparative analyses of unbiased acoustic features. We introduce methods for analyzing vocalizations as both discrete sequences and as continuous latent variables. Each method can be used to disentangle complex spectro-temporal structure and observe long-timescale organization in communication.

## 1. Introduction

Vocal communication is a common social behavior among many species, in which acoustic signals are transmitted from sender to receiver to convey information such as identity, individual fitness, or the presence of danger. Across diverse fields, a set of shared research questions seeks to uncover the structure and mechanism of vocal communication: What information is carried within signals? How are signals produced and perceived? How does the communicative transmission of information affect fitness and reproductive success? Many methods are available to address these questions quantitatively, most of which are founded on underlying principles of abstraction and characterization of 'units' in the vocal time series (Kershenbaum et al., 2016). For example, segmentation of birdsong into temporally discrete elements followed by clustering into discrete categories has played a crucial role in understanding syntactic structure in birdsong (Kershenbaum et al., 2016; Berwick et al., 2011; Sainburg et al., 2019; Katahira et al., 2013; Markowitz et al., 2013; Cody et al., 2016; Hedley, 2016; Koumura & Okanoya, 2016; Gentner & Hulse, 1998).

The characterization and abstraction of vocal communication signals remains both an art and a science. In a recent survey, Kershenbaum et. al., (2016) outline four common steps used in many analyses to abstract and describe vocal sequences: (1) the collection of data, (2) segmentation of vocalizations into units, (3) characterization of sequences, and (4) identification of meaning. A number of heuristics guide these steps, but it is largely up to the experimenter to determine which heuristics to apply and how. This application typically requires expert-level knowledge, which in turn can be difficult and time-consuming to acquire, and often unique to the structure of each species' vocal repertoire. For instance, what constitutes a 'unit' of humpback whale song? Do these units generalize to other species? Should they? When such intuitions are available they should be considered, of course, but they are generally rare in comparison to the wide range of communication signals observed naturally. As a result, communication remains understudied in most of the thousands of vocally communicating species. Even in well-documented model species, characterizations of vocalizations are often influenced by human perceptual and cognitive biases (Suzuki et al., 2006; Tyack, 1998; Janik, 1999; Kershenbaum et al., 2016). We explore a class of unsupervised, computational, machine learning techniques that avoid many of the foregoing limitations, and provide an alternative method to characterize vocal communication signals. Machine learning methods are designed to capture statistical patterns in complex datasets and have flourished in many domains (LeCun et al., 2015; Bengio et al., 2013; Radford et al., 2015; Becht et al., 2019; Brown & De Bivort, 2018; Becht et al., 2019). These techniques are therefore

[1]University of California, San Diego, USA. Correspondence to: Tim Sainburg <tsainbur@ucsd.edu>.

*Proceedings of the 37th International Conference on Machine Learning*, Vienna, Austria, PMLR 119, 2020. Copyright 2020 by the author(s).

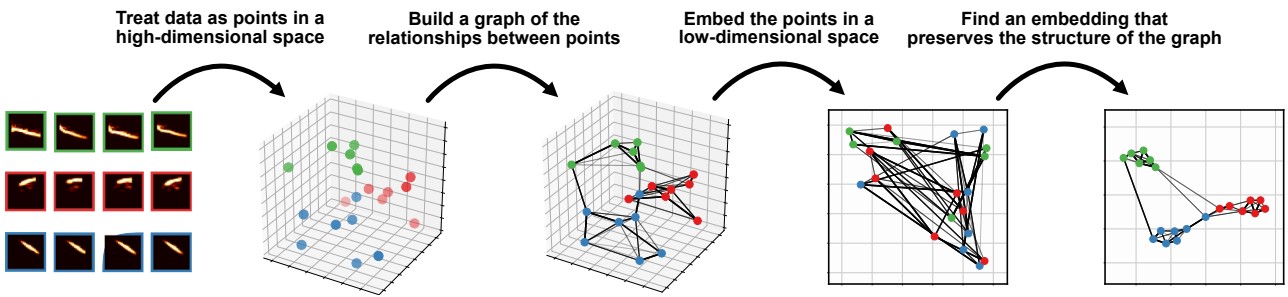

**Figure 1.** Graph-based dimensionality reduction. Current non-linear dimensionality reduction algorithms like TSNE, UMAP, and ISOMAP work by building a graph representing the relationships between high-dimensional data points, projecting those data points into a low-dimensional space, and then finds and embedding that retains the structure of the graph. This figure is for visualization, the spectrograms do not actually correspond to the points in the 3D space.

well suited to quantitatively investigate complex statistical structure in vocal repertoires that otherwise rely upon expert intuitions. In this paper, we demonstrate the utility of unsupervised latent models, statistical models that learn latent (compressed) representations of complex data, in describing animal communication.

### 1.1. Latent models of acoustic communication

Dimensionality reduction refers to the compression of high-dimensional data into a smaller number of dimensions, while retaining the structure and variance present in the original high-dimensional data. Each point in the high-dimensional input space can be projected into the lower-dimensional 'latent' feature space, and dimensions of the latent space can be thought of as features of the dataset. Animal vocalizations are good targets for dimensionality reduction. They appear naturally as sound pressure waveforms with rich, multi-dimensional temporal and spectral variations, but can generally be explained by lower-dimensional dynamics (Perl et al., 2011; Gardner et al., 2001; Arneodo et al., 2012). Dimensionality reduction, therefore, offers a way to infer a smaller set of latent dimensions (or features) that can explain much of the variance in high-dimensional vocalizations.

The common practice of developing a set of basis-features on which vocalizations can be quantitatively compared *(also called Predefined Acoustic Features, or PAFs)* is a form of dimensionality reduction and comes standard in most animal vocalization analysis software (e.g. Luscinia (Lachlan et al., 2018), Sound Analysis Pro (Tchernichovski & Mitra, 2004; Tchernichovski et al., 2000), BioSound (Elie & Theunissen, 2018), Avisoft (Specht, 2002), and Raven (Charif et al., 2010)). Birdsong, for example, is often analyzed on the basis of features such as amplitude envelope, Weiner entropy, spectral continuity, pitch, duration, and frequency modulation (Tchernichovski & Mitra, 2004; Kershenbaum

et al., 2016). Grouping elements of animal vocalizations (e.g. syllables of birdsong, mouse ultrasonic vocalizations) into abstracted discrete categories is also a form of dimensionality reduction, where each category is a single orthogonal dimension. In machine learning parlance, the process of determining the relevant features, or dimensions, of a particular dataset, is called *feature engineering*.

An attractive alternative to feature engineering is to project animal vocalizations into low-dimensional feature spaces that are determined directly from the structure of the data. Many methods for data-driven dimensionality reduction are available. PCA, for example, projects data onto a lower-dimensional surface that maximizes the variance of the projected data (Dunlop et al., 2007; Kershenbaum et al., 2016), while multidimensional scaling (MDS) projects data onto a lower-dimensional surface that maximally preserves the pairwise distances between data points. Both PCA and MDS are capable of learning manifolds that are linear or near-linear transformations of the original high-dimensional data space (Tenenbaum et al., 2000).

More recently developed graph-based methods extend dimensionality reduction to infer latent manifolds as non-linear transformations of the original high-dimensional space using ideas from topology (e.g. ISOMAP, UMAP, t-SNE; Tenenbaum et al. (2000); McInnes et al. (2018); Maaten & Hinton (2008)). Like their linear predecessors, these non-linear dimensionality reduction algorithms also try to find a low-dimensional manifold that captures variation in the higher-dimensional input data, but the graph-based methods allow the manifold to be continuously deformed, by for example stretching, twisting, and/or shrinking, in high dimensional space. These algorithms work by building a topological representation of the data and then learning a low-dimensional embedding that preserves the structure of the topological representation (Fig 1). For example, while MDS learns a low-dimensional embedding

that preserves the pairwise distance between points in Euclidean space, ISOMAP (Tenenbaum et al., 2000), one of the original topological non-linear dimensionality reduction algorithms, infers a graphical representation of the data and then performs MDS on the pairwise distances between points within the graph (geodesics) rather than in Euclidean space.

In this paper, we describe a class of nonlinear latent models that learn complex feature-spaces of vocalizations, requiring few *a priori* assumptions about the features that best describe a species' vocalizations. We show that these methods reveal informative, low-dimensional, feature-spaces that enable the formulation and testing of hypotheses about animal communication. We apply our method to diverse datasets consisting of over 20 species, including humans, bats, songbirds, mice, cetaceans, and nonhuman primates. We introduce methods for treating vocalizations both as sequences of temporally discrete elements such as syllables, as is traditional in studying animal communication (Kershenbaum et al., 2016), as well as temporally continuous trajectories, as is becoming increasingly common in representing neural sequences (Cunningham & Byron, 2014). Using both methods, we show that latent projections produce visually-intuitive and quantifiable representations that capture complex acoustic features. We show comparatively that the spectrotemporal characteristics of vocal units vary from species to species in how distributionally discrete they are and discuss the relative utility of different ways to represent different communicative signals.

## 2. Results

### 2.1. Discrete latent projections of animal vocalizations

To explore the broad utility of latent models in capturing features of vocal repertoires, we analyzed nineteen datasets consisting of 400 hours of vocalizations and over 3,000,000 discrete vocal units from 29 unique species. Each vocalization dataset was temporally segmented into discrete units (e.g. syllables, notes), either based upon segmentation boundaries provided by the dataset (where available), or using a novel dynamic-thresholding segmentation algorithm that segments syllables of vocalizations between detected pauses in the vocal stream. Each dataset was chosen because it contains large repertoires of vocalizations from relatively acoustically isolated individuals that can be cleanly separated into temporally-discrete vocal units. With each temporally discrete vocal unit we computed a spectrographic representation. We then projected the spectrograms into latent feature spaces using UMAP (e.g. Figs 2, 3). From these latent feature spaces, we analyzed datasets for classic vocal features of animal communication signals, speech features, stereotypy/clusterability, and sequential organization.

**Individual identity** Many species produce caller-specific vocalizations that facilitate the identification of individuals when other sensory cues, such as sight, are not available. The features of vocalizations facilitating individual identification vary between species. We projected identity call datasets (i.e., sets of calls thought to carry individual identity information) from four different species into UMAP latent spaces (one per species) to observe whether individual identity falls out naturally within the latent space.

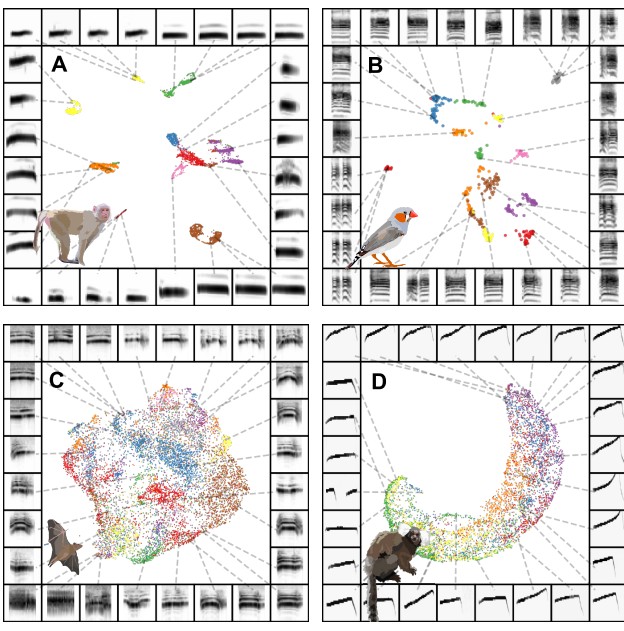

Figure 2. Individual identity is captured in projections for some datasets. Each plot shows vocal elements discretized, spectrogrammed, and then embedded into a 2D UMAP space, where each point in the scatterplot represents a single element (e.g. syllable of birdsong). Scatterplots are colored by individual identity. The borders around each plot are example spectrograms pointing toward different regions of the scatterplot. (A) Rhesus macaque coo calls. (B) Zebra finch distance calls. (C) Fruit bat infant isolation calls. (D) Marmoset phee calls.

We looked at four datasets where both caller and call-type are available. Caller identity is evident in latent projections of all four datasets (Fig 2). The first dataset is comprised of Macaque coo calls, where identity information is thought to be distributed across multiple features including fundamental frequency, duration, and Weiner entropy (Fukushima et al., 2015). Indeed, the latent projection of coo calls clustered tightly by individual identity (silhouette score = 0.378; Fig 2A). The same is true for Zebra finch distance calls (Elie & Theunissen, 2016) (silhouette score = 0.615; Fig 2B). Egyptian fruit bat pup isolation calls, which in other bat species are discriminable by adult females (Bohn et al., 2007; Engler et al., 2017; Bohn et al., 2007) clearly show regions of UMAP space densely occupied by single individ-

ual's vocalizations, but no clear clusters (silhouette score = -0.078; Fig 2C). In the marmoset phee call dataset (Miller et al., 2010) it is perhaps interesting that given the range of potential features thought to carry individual identity (Fukushima et al., 2015), phee calls appear to lie along a single continuum where each individual's calls occupy overlapping regions of the continuum (silhouette score = -0.062; Fig 2D). The silhouette score for each species was well above chance (H(2) > 20, p < $10^{-5}$). These patterns predict that some calls, such as macaque *coo* calls, would be more easily discriminable by conspecifics than other calls, such as marmoset *phee* calls.

### 2.1.1. VARIATION IN DISCRETE DISTRIBUTIONS AND STEREOTYPY

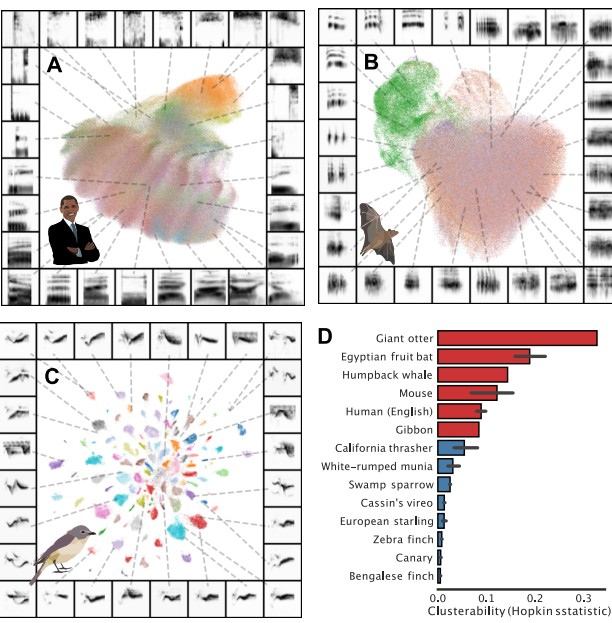

*Figure 3.* UMAP projections of vocal repertoires across diverse species. Each plot shows vocal elements segmented, spectrogrammed, and then embedded into a 2D UMAP space, where each point in the scatterplot represents a single element (e.g. syllable of birdsong). Scatterplots are colored by element categories over individual vocalizations as defined by the authors of each dataset, where available. (A) Human phonemes. (B) Egyptian fruit bat calls (color is context). (C) Cassin's vireo syllables. (O) Clusterability (Hopkin's metric) for each dataset. Lower is more clusterable. Hopkin's metric is computed over UMAP projected vocalizations for each species. Error bars show the 95% confidence interval across individuals. Color represents species category (red: mammal, blue: songbird).

In species as phylogenetically diverse as songbirds and rock hyraxes, analyzing the sequential organization of communication relies upon similar methods of segmentation and categorization of discrete vocal elements (Kershenbaum

et al., 2016). In species such as the Bengalese finch, where syllables are highly stereotyped, clustering syllables into discrete categories is a natural way to abstract song. The utility of clustering song elements in other species, however, is more contentious because discrete category boundaries are not as easily discerned (Tyack, 1998; Suzuki et al., 2006; Goffinet et al., 2019; Hertz et al., 2019).

To compare broad structural characteristics across a wide sampling of species, we projected vocalizations from 14 datasets of different species vocalizations, ranging across songbirds, cetaceans, primates, and rodents into UMAP space (Fig 3). To do so, we sampled from a diverse range of datasets, each of which was recorded from a different species in a different setting. Some datasets were recorded from single isolated individuals in a sound isolated chamber in a laboratory setting, while others were recorded from large numbers of freely behaving individuals in the wild. In addition, the units of vocalization from each dataset are variable. We used the smallest units of each vocalization that could be easily segmented, for example, syllables, notes, and phonemes. Thus, this comparison across species is not well-controlled. Still, such a dataset enabling a broad comparison in a well-controlled manner does not exist. Latent projections of such diverse recordings, while limited in a number of ways, have the potential to provide a glimpse into broad structure into vocal repertoires, yielding novel insights into broad trends in animal communication. For each dataset, we computed spectrograms of isolated elements, and projected those spectrograms into UMAP space (Fig 3). Where putative element labels are available, we plot them in color over each dataset.

Visually inspecting the latent projections of vocalizations reveals appreciable variability in how the repertoires of different species cluster in latent space. For example, mouse USVs appear as a single cluster (Fig 3I), while zebra finch syllables appear as multiple discrete clusters (Fig 3M,F), and gibbon song sits somewhere in between (Fig 3L). This suggests that the spectro-temporal acoustic diversity of vocal repertoires fall along a continuum ranging from unclustered and uni-modal to highly clustered.

We quantified this effect using a linear mixed-effects model comparing the Hopkin's statistic across UMAP projections of vocalizations from single individuals (n = 289), controlling for the number of vocalizations produced by each individual as well as random variability at the level of species. We included each of the species in Fig 3 except giant otter and gibbon vocalizations, as individual identity was not available for those datasets. We find that songbird vocalizations are significantly more clustered than mammalian vocalizations ($\chi^2(1) = 20$, p < $10^{-5}$). The stereotypy of songbird (and other avian) vocal elements is well documented (Williams, 2004; Smith et al., 1997) and at least in

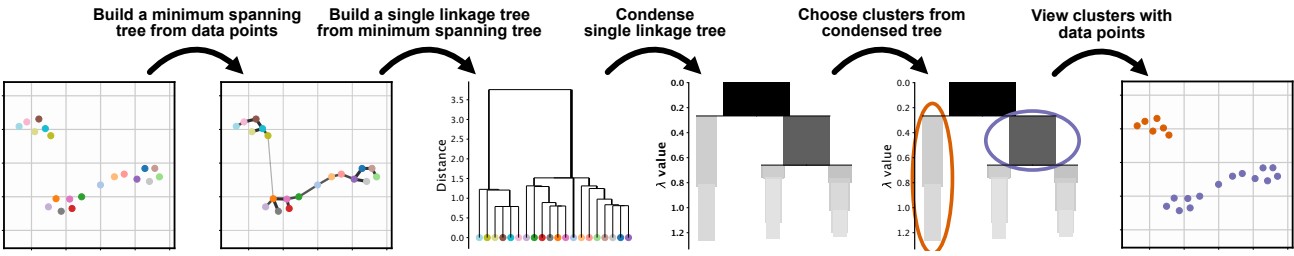

*Figure 4.* HDBSCAN density-based clustering. Clusters are found by generating a graphical representation of data, and then clustering on the graph. The data shown in this figure are from the latent projections from Fig 1. Notably, the three clusters in Fig 1. are clustered into only two clusters using HDBSCAN, exhibiting a potential shortcoming of the HDBSCAN algorithm. The grey colormap in the condensed trees represent the number of points in the branch of the tree. Λ is a value used to compute the persistence of clusters in the condensed trees.

zebra finches is related to the high temporal precision in the singing-related neural activity of vocal-motor brain regions (Hahnloser et al., 2002; Fee et al., 2004; Chi & Margoliash, 2001).

### 2.1.2. CLUSTERING VOCAL ELEMENT CATEGORIES

UMAP projections of birdsongs largely fall more neatly into discriminable clusters (Fig 3). If clusters in latent space are highly similar to experimenter-labeled element categories, unsupervised latent clustering could provide an automated and less time-intensive alternative to hand-labeling elements of vocalizations. To examine this, we compared how well clusters in latent space correspond to experimenter-labeled categories in three human-labeled datasets: two separate Bengalese finch datasets (Nicholson et al., 2017; Koumura, 2016), and one Cassin's vireo dataset (Hedley, 2016). We compared four different labeling techniques: a hierarchical density-based clustering algorithm (HDBSCAN; (Campello et al., 2013; McInnes et al., 2017)) applied to UMAP projections of spectrograms, HDBSCAN applied to PCA projections of spectrograms[1], k-means (Pedregosa et al., 2011) clustering applied over UMAP, and k-means clustering applied over spectrograms. We found that HDBSCAN clustering outperformed other clustering algorithms on all metrics for all datasets (See full manuscript).

Like the contrast between MDS and UMAP, the k-means clustering algorithm works directly on the Euclidean distances between data points, whereas HDBSCAN operates on a graph-based transform of the input data (Fig 4). Briefly, HDBSCAN first defines a 'mutual reachability' distance between elements, a measure of the distance between points in the dataset weighted by the local sparsity/density of each point (measured as the distance to a $k$th nearest neighbor). HDBSCAN then builds a graph, where each edge between

vertices (points in the dataset) is the mutual reachability between those points, and then prunes the edges to construct a minimum spanning tree (a graph containing the minimum set of edges needed to connect all of the vertices). The minimum spanning tree is converted into a hierarchy of clusters of points sorted by mutual reachability distance, and then condensed iteratively into a smaller hierarchy of putative clusters. Finally, clusters are chosen as those that persist and are stable over the greatest range in the hierarchy.

### 2.1.3. ABSTRACTING AND VISUALIZING SEQUENTIAL ORGANIZATION

As acoustic signals, animal vocalizations have an inherent temporal structure that can extend across time scales from short easily discretized elements such as notes, to longer duration syllables, phrases, songs, bouts, etc. The latent projection methods described above can be used to abstract corpora of song elements well-suited to temporal pattern analyses (Sainburg et al., 2019), and to make more direct measures of continuous vocalization time series. Moreover, their automaticity enables the high throughput necessary to satisfy intensive data requirements for most quantitative sequence models.

In practice, modeling sequential organization can be applied to any discrete dataset of vocal elements, whether labeled by hand or algorithmically. Latent projections of vocal element have the added benefit of allowing visualization of the sequential organization that can be compared to abstracted models. As an example of this, we derived a corpus of symbolically segmented vocalizations from a dataset of Bengalese finch song using latent projections and clustering (Fig 5). Bengalese finch song bouts comprise a small number (~5-15) of highly stereotyped syllables produced in well-defined temporal sequences a few dozen syllables long (Katahira et al., 2013). We first projected syllables from a single Bengalese finch into UMAP latent space, then visualized transitions between vocal elements

---

[1]HDBSCAN is applied to 100-dimensional PCA projections rather than spectrograms directly because HDBSCAN does not perform well in high-dimensional spaces (McInnes et al., 2017).

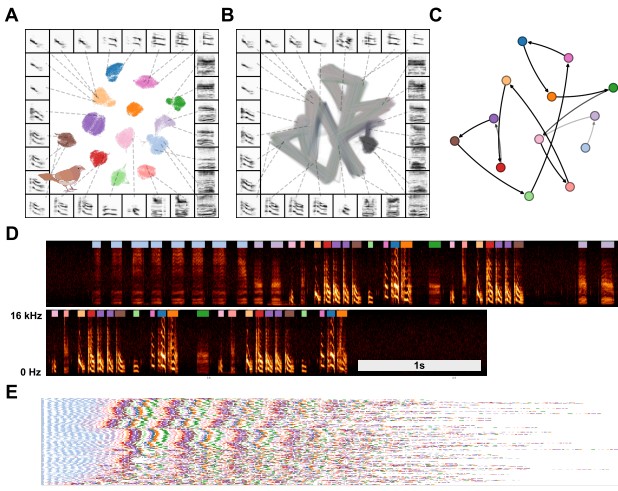

*Figure 5.* Latent visualizations of Bengalese finch song sequences. (A) Syllables of Bengalese finch songs from one individual are projected into 2D UMAP latent space and clustered using HDB-SCAN. (B) Transitions between elements of song are visualized as line segments, where the color of the line segment represents its position within a bout. (C) The syllable categories and transitions in (A) and (B) can be abstracted to transition probabilities between syllable categories, as in a Markov model. (D) An example vocalization from the same individual, with syllable clusters from (A) shown above each syllable. (E) A series of song bouts. Each row is one bout, showing overlapping structure in syllable sequences. Bouts are sorted by similarity to help show structure in song.

in latent space as line segments between points (Fig 5B), revealing highly regular patterns. To abstract this organization to a grammatical model, we clustered latent projections into discrete categories using HDBSCAN. Each bout is then treated as a sequence of symbolically labeled syllables (e.g. $B \rightarrow B \rightarrow C \rightarrow A$; Fig 5D) and the entire dataset rendered as a corpus of transcribed song (Fig 5E). Using the transcribed corpus, one can abstract statistical and grammatical models of song, such as the Markov model shown in Fig 5C or the information-theoretic analysis in Sainburg et al., (2019).

## 2.2. Temporally continuous latent trajectories

Not all vocal repertoires are made up of elements that fall into highly discrete clusters in latent space (Fig 3). For several of the datasets we analysed, categorically discrete elements are not readily apparent, making analyses such as the cluster-based analyses performed in Figure 5 more difficult. In addition, many vocalizations are difficult to segment temporally, and determining what features to use for segmentation requires careful consideration (Kershenbaum et al., 2016). In many bird songs, for example, clear pauses exist between song elements that enable one to distinguish sylla-

bles. In other vocalizations, however, experimenters must rely on less well-defined physical features for segmentation (Janik, 1999; Kershenbaum et al., 2016), which may in turn invoke a range of biases and unwarranted assumptions. At the same time, much of the research on animal vocal production, perception, and sequential organization relies on identifying "units" of a vocal repertoire (Kershenbaum et al., 2016). To better understand the effects of temporal discretization and categorical segmentation in our analyses, we considered vocalizations as continuous trajectories in latent space and compared the resulting representations to those that treat vocal segments as single points (as in the previous Bengalese finch example in Fig 5). We show here explorations of two datasets: Bengalese finch (Fig 6) and human speech (Fig 7). In both dataset, we find that continuous latent trajectories capture short and long timescale structure in vocal sequences without requiring vocal elements to be segmented or labeled.

### 2.2.1. COMPARING DISCRETE AND CONTINUOUS REPRESENTATIONS OF SONG IN THE BENGALESE FINCH

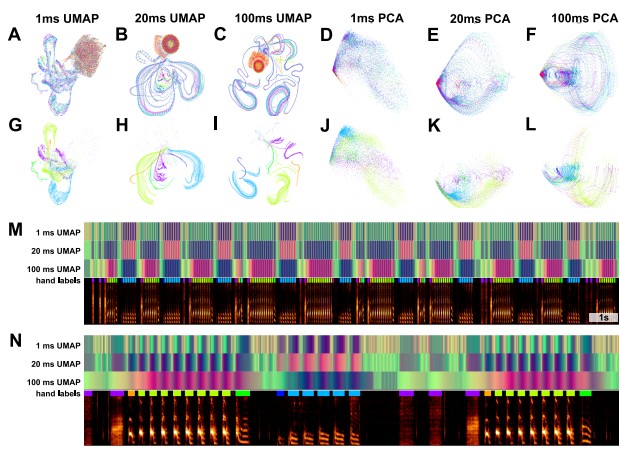

*Figure 6.* Continuous UMAP projections of Bengalese finch song from a single bout produced by one individual. (A-C) Bengalese finch song is segmented into either 1ms (A), 20ms (B), or 100ms (C) rolling windows of song, which are projected into UMAP. Color represents time within the bout of song. (D-F) The same plots as in (A), projected into PCA instead of UMAP. (G-I) The same plots as (A-C) colored by hand-labeled element categories (unlabelled points are not shown). (J-L) The same plot as (D-E) colored by hand-labeled syllable categories. (M) UMAP projections represented in colorspace over a bout spectrogram. The top three rows are the UMAP projections from (A-C) projected into RGB colorspace to show the position within UMAP space over time as over the underlying spectrogram data. The fourth row are the hand labels. The final row is a bout spectrogram. (N) a subset of the bout shown in (M). In G-L, unlabeled points (points that are in between syllables) are not shown for visual clarity.

Bengalese finch song provides a relatively easy visual comparison between the discrete and continuous treatments of song, because it consists of a small number of unique highly stereotyped syllables (Fig 6). With a single bout of Bengalese finch song, which contains several dozen syllables, we generated a latent trajectory of song as UMAP projections of temporally-rolling windows of the bout spectrogram (See Projections section). To explore this latent space, we varied the window length between 1 and 100ms (Fig 6A-L). At each window size, we compared UMAP projections (Fig 6A-C) to PCA projections (Fig 6D-F). In both PCA and UMAP, trajectories are more clearly visible as window size increases across the range tested, and overall the UMAP trajectories show more well-defined structure than the PCA trajectories. To compare continuous projections to discrete syllables, we re-colored the continuous trajectories by the discrete syllable labels obtained from the dataset. Again, as the window size increases, each syllable converges to a more distinct trajectory in UMAP space (Fig 6G-I). To visualize the discrete syllable labels and the continuous latent projections in relation to song, we converted the 2D projections into colorspace and show them as a continuous trajectory alongside the song spectrograms and discrete labels in Figure 6M,N. Colorspace representations of the 2D projections consist of treating the two UMAP dimensions as either a red, green, or blue channel in RGB (3D) colorspace, and holding the third channel constant. This creates a colormap projection of the two UMAP dimensions.

### 2.2.2. LATENT TRAJECTORIES OF HUMAN SPEECH

Discrete elements of human speech (i.e. phonemes) are not spoken in isolation and their acoustics are influenced by neighboring sounds, a process termed co-articulation. For example, when producing the words 'day', 'say', or 'way', the position of the tongue, lips, and teeth differ dramatically at the beginning of the phoneme 'ey' due to the preceding 'd', 's', or 'w' phonemes, respectively. This results in differences in the pronunciation of 'ey' across words (Fig 7E). Co-articulation explains much of the acoustic variation observed within phonetic categories. Abstracting to phonetic categories therefore discounts much of this context-dependent acoustic variance.

We explored co-articulation in speech, by projecting sets of words differing by a single phoneme (i.e. minimal pairs) into continuous latent spaces, then extracted trajectories of words and phonemes that capture sub-phonetic context-dependency (Fig 7). We obtained the words from the Buckeye corpus of conversational English. We computed spectrograms over all examples of each target word, then projected sliding 4-ms windows from each spectrogram into UMAP latent space to yield a continuous vocal trajectory over each word (Fig 7). We visualized trajectories by their corresponding word and phoneme labels (Fig 7A,B) and computed the

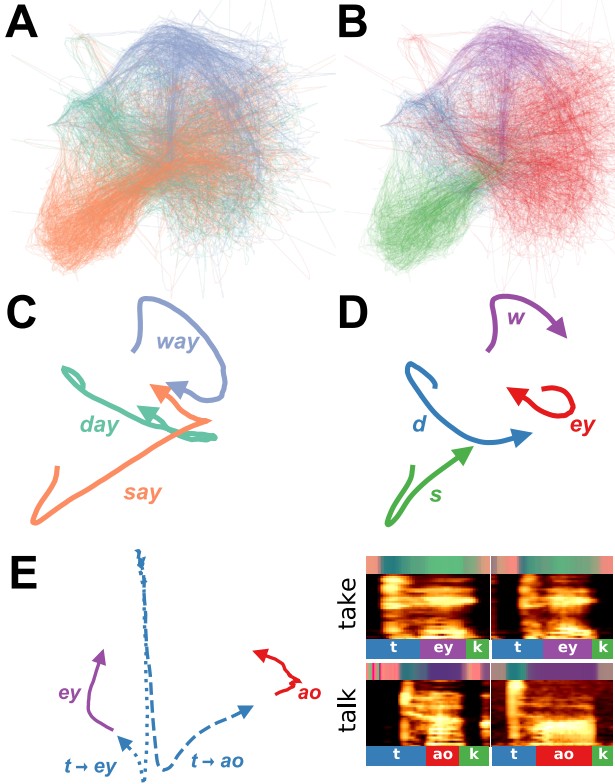

Figure 7. Speech trajectories showing coarticulation in minimal pairs. (A) Utterances of the words 'day', 'say', and 'way' are projected into a continuous UMAP latent space with a window size of 4ms. Color represents the corresponding word. (B) The same projections are colored by the corresponding phonemes. (D) The average latent trajectory for each word. (E) The average trajectory for each phoneme. (F) Example spectrograms of words, with latent trajectories above spectrograms and phoneme labels below spectrograms. (G) Average trajectories and corresponding spectrograms for the words 'take' and 'talk' showing the different trajectories for 't' in each word. (H) Average trajectories and the corresponding spectrograms for the words 'then' and 'them' showing the different trajectories for 'eh' in each word.

average latent trajectory for each word and phoneme (Fig 7C,D). The average trajectories reveal context-dependent variation within phonemes caused by coarticulation. For example, the words 'way', 'day', and 'say' each end in the same phoneme ('ey'; Fig 7A-D), which appears as an overlapping region in the latent space (the red region in Fig 7C). The endings of each average word trajectory vary, however, indicating that the production of 'ey' differs based on its specific context (Fig 7C). The difference between the production of 'ey' can be observed in the average latent trajectory over each word, where the trajectories for 'day' and 'say' end in a sharp transition, while the trajectory for 'way' is more smooth (Fig 7C). Latent space trajectories can

reveal other co-articulations as well. In Figure 7E, we show the different trajectories characterizing the phoneme 't' in the context of the word 'take' versus 'talk'. In this case, the 't' phoneme follows a similar trajectory for both words until it nears the next phoneme ('ey' vs. 'ao'), at which point the production of 't' diverges for the different words.

## 3. Discussion

We have presented a set of computational methods for projecting vocal communication signals into low-dimensional latent representational spaces, learned directly from the spectrograms of the signals. We demonstrate the flexibility and power of these methods by applying them to a wide sample of animal vocal communication signals, including songbirds, primates, rodents, bats, and cetaceans (Fig 3). Deployed over short timescales of a few hundred milliseconds, our methods capture significant behaviorally-relevant structure in the spectro-temporal acoustics of these diverse species' vocalizations. We find that complex attributes of vocal signals, such as individual identity (Fig 2), species identity, geographic population variability, phonetics, and similarity-based clusters can all be captured by the unsupervised latent space representations we present. We also show that songbirds tend to produce signals that cluster discretely in latent space, whereas mammalian vocalizations are more uniformly distributed, an observation that deserves much closer investigation in more species. Applied to longer timescales, spanning seconds or minutes, the same methods allowed us to visualize sequential organization and test models of vocal sequencing (Fig 5). We demonstrated that in some cases latent approaches confer advantages over hand labeling or supervised learning (See full manuscript/code). Finally, we visualized vocalizations as continuous trajectories in latent space (Figs 6, 7), providing a powerful method for studying sequential organization without discretization (Kershenbaum et al., 2016).

Latent models have shown increasing utility in the biological sciences over the past several years. As machine learning algorithms improve, so will their utility in characterizing the complex patterns present in biological systems like animal communication. In neuroscience, latent models already play an important role in characterizing complex neural population dynamics (Cunningham & Byron, 2014). Similarly, latent models are playing an increasingly important role in computational ethology (Brown & De Bivort, 2018), where characterizations of animal movements and behaviors have uncovered complex sequential organization (Marques et al., 2018; Berman et al., 2016; Wiltschko et al., 2015). In animal communication, pattern recognition using various machine learning techniques has been used to characterize vocalizations and label auditory objects (Sainburg et al., 2019; Cohen et al., 2019; Coffey et al., 2019; Van Segbroeck et al.,

2017; Goffinet et al., 2019; Kollmorgen et al., 2019; Hertz et al., 2019). Our work furthers this emerging research area by demonstrating the utility of unsupervised latent models for both systematically visualizing and abstracting structure from animal vocalizations across a wide range of species.

## Software and Data

All software is publicly available and example Jupyter Notebooks are provided for each species vocal repertoire and analyses type (`https://github.com/timsainb/avgn_paper`). The data is provided in Supplementary Table 1 of the longform paper.

## Acknowledgements

Work supported by NSF GRF 2017216247 and an Annette Merle-Smith Fellowship to T.S. and NIH DC0164081 and DC018055 to T.Q.G. We additionally would like to thank Kyle McDonald and his colleagues for motivating some of our visualization techniques with their work on humpback whale song (McDonald, 2019).

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
