# OpenReview forum: "Finding, visualizing, and quantifying latent structure across diverse animal vocal repertoires"
_ICML.cc/2020/Workshop/SAS — Submitted to SAS 2020_

### Official Review · AnonReviewer4 · 2020-06-30
**A detailed analysis of animal sound recordings**

**Confidence:** 3
**Rating:** 5

**Review:**

This paper analyses animal sounds using different visualization techniques. I found this paper interesting and very well-written. The results reported are analyzed in detail.  However, the paper is focusing on visualization rather than on standard self-supervised learning. Moreover, I don't see novelty from the pure machine learning point of view. The work, in fact, uses well-known visualization (e.g, UMAP) and clustering techniques only.  Given the addressed topic, I think this contribution would be more appropriate in another venue where the impressive analysis of the results could be better appreciated.

Minor comment:
The paper is 8 pages long against the 5 pages allowed.

---

### Official Review · AnonReviewer5 · 2020-07-01
**Latent sub-spaces of animal vocalisations.**

**Confidence:** 3
**Rating:** 6

**Review:**

This work investigates different non-linear projections to learn low dimensional sub-spaces to describe various species vocalisation. Two classes of techniques operating either on discrete latent projections or continuous ones are used to visualise and cluster numerous attributes characterising 20 species vocalisations.

First of all, the paper is very well written and self-contained. It is also very interesting to have such an in depth analysis of a less common research area (animals vocalisation). More precisely, this paper offers tons of insights on how "speech" signals coming from various animals may characterise a species or a single animal in this species. Furthermore, a case study is also given on Human speech, allowing for further understanding of the major differences existing between Humans and others animals.

My main concern is the adéquation of this paper to the topic of the workshop. Sure, all this work is "unsupervised" or "weakly-supervised". Nonetheless, I don't see any clear link going from the latent representations that are obtained and a potential use with machine learning. Of course, it is easy to think of many, but the paper should have offered a discussion, and some experiments going into this direction to be aligned with the goal of the workshop. As an example, one of these techniques could have been used to derive better features on a "Speaker identification" system for Rhesus macaque (that seems to be well-discriminated). Representation learning is part of the topics of interest, but I still think that a link with machine learning is missing somewhere.

Overall, I think this is a very important and under appreciated line of work, and this paper is very clear and would surely provide tons of valuable content for researchers interested in this domain.

---

### Official Review · AnonReviewer1 · 2020-07-01
**A very well written paper with in-depth analysis**

**Confidence:** 3
**Rating:** 6

**Review:**

The paper is very well written. It describes a latent model for studying complex audio data. It does not use any recently developed discrete representation learning techniques like VQ-VAE (or any other deep learning based dimensionality reduction or representation learning techniques). Representation coming from a model trained on AudioSet would be a good baseline for this technique to compare against.

Overall, the paper seems to be a valuable contribution and will be interesting to the readers.

---

### Decision · Program_Chairs · 2020-07-01

**Decision:**

Reject

**Comment:**

Dear author(s),

Thank you very much for your submission at the ICML2020@SaS workshop (https://icml-sas.gitlab.io/). Based on the scores assigned by the reviewers, we regret to inform you that the paper was rejected. We got 26 submissions and we were only able to accept 13 papers.
We found the paper interesting and well-written, but a bit out-of-topic for the workshop. We believe that this contribution could be better appreciated in another venue.
We invite you anyway to consider the feedback of the reviewers and to follow our upcoming workshop on July 17.